# The effect of the COVID-19 pandemic on delirium incidence in Ontario long-term care homes: A retrospective cohort study

**Lydia Kennedy[1]◉, John P. Hirdes[2]◉, George Heckman[2]◉, Samuel D. Searle[1]◉, Caitlin McArthur◉[3]◉ ***

**1** Department of Medicine, Dalhousie University, Halifax, Nova Scotia, Canada, **2** School of Public Health Sciences, University of Waterloo, Waterloo, Ontario, Canada, **3** Department of Physiotherapy, Dalhousie University, Halifax, Nova Scotia, Canada

◉ These authors contributed equally to this work.
* caitlin.mcarthur@dal.ca

**Data Availability Statement:** Data from the Continuing Care Reporting System is owned by the Canadian Institute for Health Information and provided in deidentified, encrypted form to the

## Abstract

### Objectives

To describe delirium incidence before and during the COVID-19 pandemic and examine factors associated with delirium incidence in the long-term care setting.

### Methods

We conducted a retrospective cohort study of Ontario long-term care residents without severe cognitive impairment or baseline delirium with an assessment between February 1, 2019, and March 31, 2021. Data were collected from the interRAI Minimum Data Set (MDS) 2.0. The outcome of interest was delirium development. Selected independent variables were entered into univariate longitudinal generalized estimating equations, followed by multivariate analysis. Odds ratios (ORs) and 95% confidence intervals (CIs) are reported.

### Results

A total of 63,913 residents were included within the comparison sample from February 2019 to February 2020. The pandemic sample consisted of 54,867 residents from March 2020 to March 2021. Incidence of delirium in the comparison sample was 3.4% (2158 residents) compared to 3.2% (1746 residents) in the pandemic sample (*P* = 0.06). Residents who were older, cognitively impaired, and increasingly frail had greater odds of developing delirium. Increased odds were associated with a diagnosis of bipolar disorder (OR 1.27, 95% CI 1.07–1.51) and anxiolytic use (OR 1.12, 95% CI 1.01–1.25). Residents who were newly admitted (OR 0.65, 95% CI 0.60–0.71) and those dependent for activities of daily living (OR 0.46, 95% CI 0.33–0.64) had lower odds of delirium development.

### Conclusions and implications

The incidence of delirium did not differ between the year prior to and the first year of the COVID-19 pandemic, indicating that preventative interventions employed by long-term care

University of Waterloo. No other sites are permitted to receive these data under this data sharing agreement. Access to the Continuing Care Reporting System may be sought upon reasonable and justifiable request from the Canadian Institute for Health Information. Data requests may be made using CIHI's Data Request Form: https://www.cihi.ca/en/access-data-andreports/make-a-data-request.

**Funding:** The study is supported by funding from the Government of Canada's New Frontiers in Research Fund (NFRF; NFRFG-2020-00500) for collaboration in the EU Horizon 2020 research and innovation project Individualized CARE for Older Persons with Complex Chronic Conditions in Home Care and Nursing Homes (I-CARE4OLD, Grant Agreement No 965341). In addition, it received funding through the Canadian Institutes of Health Research (CIHR Reference # GA6-177780). There was no additional external funding received for this study. The funders had no role in study design, data collection and analysis, decision to publish, or preparation of the manuscript.

**Competing interests:** The authors have declared that no competing interests exist.

homes may have been effective. Long-term care residents who are older, frail, cognitively impaired, or had unstable health would benefit from targeted interventions to prevent delirium. Newly admitted residents or those dependent in activities of daily living had lower odds of developing delirium, which could indicate under detection in these groups.

## Introduction

Delirium is a condition characterized by an acute change in mental status where an individual's attention and awareness fluctuate throughout the day [1]. Delirium is accompanied by disturbances in cognition and represents a change from the individual's baseline cognitive function [1]. Predisposing risk factors for delirium include increased age, pre-existing cognitive impairment, frailty, psychiatric illness, poor nutritional status, and visual or hearing impairment [2]. Delirium can develop in the presence of several different precipitating factors such as acute medical illness, trauma, surgery, dehydration, and psychological stress [3]. The condition can lead to hospitalization and loss of independence, is associated with an increased risk of further cognitive decline and dementia, and has been associated with significantly increased morbidity and mortality [3, 4]. Additionally, it has been estimated that between 30 to 40% of delirium cases are preventable [3].

Many long-term care (LTC) residents have an increased prevalence of these risk factors and consequently have been found to be at an increased risk of delirium relative to individuals living in the community [5]. Numerous studies have looked at LTC homes in Canada to estimate the prevalence of delirium among this population. Results are wide ranging, with prevalence between 3.6–40.4% [6–8]. An international review looked more widely at studies conducted across the world, finding a prevalence ranging from 1.4 to 70.3% depending on the structure and individual characteristics of the home [5].

On March 11, 2020, the World Health Organization characterized COVID-19 as a global pandemic [9]. Following this, on March 17, 2020, Ontario, the largest province in Canada, declared a provincial state of emergency [10]. Worldwide, the COVID-19 pandemic led to an abundance of new infection control measures and visitation protocols in LTC homes. Many of these new measures were initiated to safeguard the health of LTC residents and limit spread of the virus. However, these measures were hypothesized to restrict the usage of evidence-based delirium prevention strategies. Residents experienced increased isolation and reduced family presence, which is a vital part of reorientation and family-centred care to prevent delirium [2]. Staffing limitations restricted the ability for frequent mobilization, and the usage of personal protective equipment, such as masks and face shields, caused communication barriers for those with hearing or vision impairment [11, 12]. Due to these factors, it could be hypothesized that residents were more likely to develop delirium during the pandemic than before. However, much remains unknown about the effect of the pandemic on delirium incidence in LTC. Much of the research regarding delirium during the pandemic has focused on the in-patient hospital setting [13–16]. Delirium incidence and its associated factors in LTC homes during the COVID-19 pandemic has not been fully investigated. The Canadian Institute for Health Information found a 36% decrease in the number of LTC residents transferred to hospital due to delirium in 2020 compared to 2019 [17]. While it seems that this may reflect a decrease in delirium occurrence in LTC homes during the pandemic, the actual incidence of delirium was not described. It is unknown whether there was a true decrease in delirium, or whether

residents developing delirium were instead kept at the LTC home to prevent possible infection with COVID-19 in the hospital setting.

Therefore, our study aims to describe delirium incidence in Ontario LTC homes before and during the COVID-19 pandemic, and the associations between incidence and various clinical factors. Understanding factors that lead to delirium development is key because prevention is critical to minimize incidence. This is especially pertinent in the setting of COVID-19 as evaluating the extent of delirium in LTC homes during the pandemic can aid us in understanding whether preventative strategies decreased incidence and help us prepare in the event of another pandemic.

## Methods

### Study design and data sources

The data obtained for this retrospective study was from the interRAI Minimum Data Set (MDS) 2.0 in Ontario, Canada. This data is submitted to the Canadian Institute for Health Information, anonymized, and then held on a remote server at the University of Waterloo. The use of this de-identified data for secondary analysis was approved by the Health Sciences Research Ethics Board at Dalhousie University (REB#2022–6106). Data was accessed beginning from May 7, 2022.

The MDS 2.0 is a valid and reliable standardized assessment tool completed by trained assessors in each LTC home [18]. It collects sociodemographic and clinical information from the resident, their family, and through their medical records. The assessment is completed upon an individual's admission to a LTC home, at quarterly and yearly intervals within the home, and any time there is a change in the health status of the LTC resident. The MDS 2.0 information is used within LTC homes to help determine care needs and support care planning [18]. Throughout the COVID-19 pandemic, assessments continued to be performed regularly in LTC homes in Ontario.

### Sample construction

The pre-pandemic comparison sample included all LTC residents with an admission, annual, or quarterly assessment completed between February 1, 2019, and February 29, 2020. The first assessment performed within this period was selected as the baseline assessment for the resident. Admission and annual assessments are comprehensive and include all of the variables required for our study. However, quarterly assessments are shorter to complete and do not include certain items that are not expected to change acutely. Therefore, for a subset of each sample where a quarterly assessment was used as a resident's baseline, certain independent variables of interest were not carried forward. These were therefore obtained from the most recent admission or annual assessment completed in the preceding year. As noted, variables carried forward included those expected to remain fairly constant, for example, marital status, hearing/vision impairment, Alzheimer's status, and a subset of variables used within the Frailty Index. The variable used to determine development of delirium is captured within these quarterly assessments and did not need to be carried forward. All subsequent annual or quarterly assessments that fell within the comparison period were used as follow-ups. The pandemic sample was constructed in the same manner, including all LTC residents with an admission, annual, or quarterly assessment completed between March 1, 2020, and March 31, 2021.

Residents whose discharge was projected to be within 90 days, those receiving hospice care, and/or residents with end stage disease defined as 6 months or less to live were excluded. The Cognitive Performance Scale (CPS) assesses a resident's cognitive status with scores ranging from 0 (no cognitive impairment) to 6 (very severe cognitive impairment) [19]. Residents with

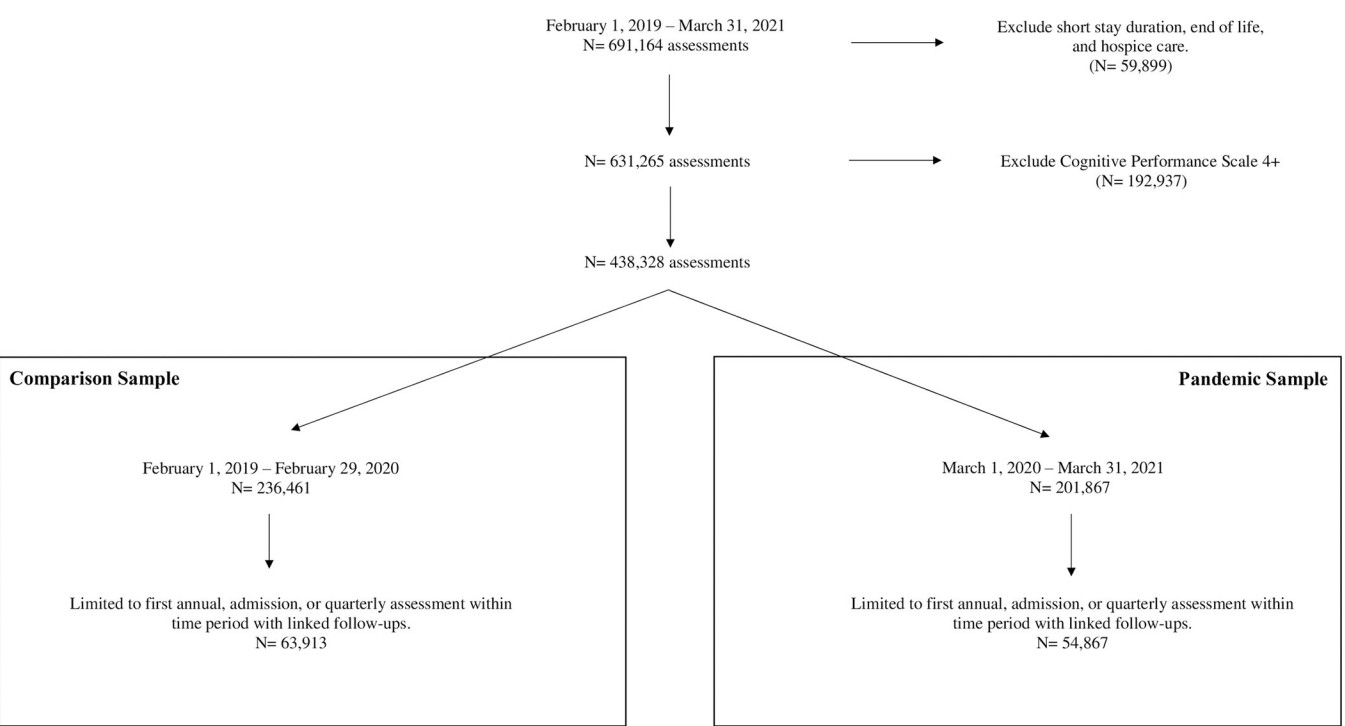

**Fig 1. Comparison and pandemic sample creation.** All LTC residents with an assessment between February 1, 2019 and March 31, 2021 were compiled. Those who were receiving hospice care, had a life expectancy of less than 6 months, or who had a projected discharge within 90 days were excluded. Residents with a Cognitive Performance Scale score of greater than 4 were excluded. Those with delirium at their baseline assessment were excluded. The comparison sample was constructed of annual, admission, or quarterly assessments completed between February 1, 2019 and February 29, 2020. The pandemic sample was constructed of annual, admission, or quarterly assessments completed between March 1, 2020, and March 31, 2021.

a score of 4 or higher on the CPS were excluded, as it can be difficult to identify delirium in the setting of severe cognitive impairment [20]. As the aim of this study was to investigate the development of new delirium from baseline to follow-up, residents with indicators of delirium on their baseline assessment were excluded. Residents who remained in LTC throughout the period of study between February 1, 2019, and March 31, 2021, may have been included in both the comparison and pandemic samples. However, as residents with baseline delirium were deleted in both samples, it was determined that this would not have a large effect on the final incidences obtained. The construction of the comparison and pandemic samples is outlined in Fig 1.

## Variables of interest

The outcome of interest was the development of delirium. This was determined using the delirium clinical assessment protocol (CAP) within MDS 2.0. One of the more widely recognized delirium screening tools is the Confusion Assessment Method (CAM). This has been found in studies to have a high sensitivity (94%-100%) and inter-rater reliability [20]. It is based on the presence of four main symptoms: acute onset with fluctuating course, inattention, and disorganized thinking or altered level of consciousness [20]. Comparatively, the delirium CAP within MDS 2.0 looks at whether a resident is experiencing an acute change in mental status, or their behavior appears different than their usual functioning. It assesses the following behaviors: easily distracted, disorganized speech, periods of altered perception, restlessness, lethargy, or a variation in mental function throughout the day [8]. The delirium CAP is coded as triggered if the resident has experienced at least one of these symptoms in the previous

seven days. At this time, no published studies have been found assessing the comparative sensitivity and specificity of the delirium CAP in relation to the CAM. As the CAP is used as a screening tool, it is likely to have a high sensitivity and should always initiate further work-up for a more definitive diagnosis. In this study, if the CAP was triggered, the resident was defined as having developed delirium. Therefore, delirium development was dichotomous for this study.

The follow-ups established in each sample were used to determine development of delirium from baseline. If delirium occurred, we calculated the time in days between the baseline assessment and delirium development. Within the pandemic sample, we also assessed the date of delirium development and categorized this relative to waves of the COVID-19 pandemic. We defined the waves of the pandemic based on the weekly number of COVID-19 cases reported by the Government of Canada [21]: wave 1 from March 1, 2020 to June 30, 2020 and waves 2 and 3 overlapping from October 1, 2020 to March 31, 2021. July 1, 2020 to September 30, 2020 was defined as off-peak as COVID-19 cases in Canada decreased during this time.

The independent variables were chosen based on previous literature looking at predisposing and precipitating factors of delirium. Baseline assessments provided demographic data including age, sex, and marital status. A diagnosis of dementia [2, 22, 23], including Alzheimer's, along with numerous psychiatric disorders (anxiety, bipolar, depression, schizophrenia) [24–27] were chosen. Hearing impairment [2, 28], vision impairment [2, 28, 29], presence of dehydration [30], and occurrence of a hospital stay within the last 90 days were included [31]. Weight loss was categorized as occurring if there had been a loss of greater than 5% in the past 30 days or greater than 10% in the past 180 days [32, 33]. Numerous variables associated with medication use were chosen: number of medications being taken [33], type of medication taken in the last seven days (analgesics, antidepressants, antipsychotics, anxiolytics, diuretics, hypnotics) [31, 34], and whether there had been start of a new medication within the last 90 days [34]. Several validated and reliable scales provided within the MDS 2.0 were incorporated to capture activities of daily living (ADL) performance [35], degree of health instability [8], cognitive status [33], and frailty [36]. The ADL Hierarchy Scale was used to determine an individual's degree of dependence in their ADLs and is scored from 0 (independent) to 6 (total dependence) [37]. The Changes in Health, End-Stage Disease and Signs and Symptoms (CHESS) scale was included to look at an individual's level of frailty-related health instability [38]. This scale scores an individual from 0 (no health instability) to 5 (severe health instability). The CPS describes cognitive impairment in a range from 0 to 6 [19]. As mentioned previously, this study excluded assessments with severe cognitive impairment, represented with a CPS score of 4 or greater. Frailty was assessed using the Frailty Index (FI), which captures a total of 58 health deficits [39]. Because this study used the delirium CAP within the MDS 2.0, variables that were used within the CAP and the FI were removed from the FI including: easily distracted, periods of altered perception or awareness of surroundings, episodes of disorganized speech, periods of restlessness, periods of lethargy, and mental function varying throughout day. Finally, baseline assessments that were categorized as an admission assessment were flagged as new admissions. These new admissions were then further assessed by looking at where the resident was admitted from (community, hospital, inpatient unit).

## Data analysis

The baseline characteristics of both the comparison and pandemic samples were expressed in count and percent. The $\chi^2$ statistic was used to determine statistically significant differences between the samples. An alpha value of 0.05 was used. The selected independent variables were entered into univariate longitudinal generalized estimating equations for the outcome of

interest, delirium development. We also used these independent variables and entered them in an interaction with the pandemic variable for the outcome of interest. Multivariate analysis was then performed by adding each of the variables to the model and retaining those that were significant at p<0.05. Odds ratios (ORs) and the 95% confidence intervals (CIs) are reported. Data analysis for this study was completed using SAS v 9.4.

## Results

A total of 227,380 assessments representing 63,913 residents were included within the comparison sample. The mean number of follow-ups within this cohort was 4, with a maximum of 7. A total of 194,419 assessments representing 54,867 residents were included within the pandemic sample, with a mean number of 4 follow-ups and a maximum of 8. The comparison sample had a mean age of 83.4 years (standard deviation (SD) = 10.6) while the pandemic sample had a mean age of 83.1 years (SD = 10.9). Both samples were predominantly female (greater than 65%). As displayed in Fig 2, the comparison sample had 2158 residents (3.4%) develop delirium, compared to the pandemic sample where 1746 residents (3.2%) developed delirium (*P* = 0.06). The mean number of days to development of delirium in the comparison sample was 167 (SD = 88). This was only 3 days greater in the pandemic sample with a mean number of days to development of 170 (SD = 89). The percentage of residents who triggered the delirium CAP was greatest in the off-peak period at 10.9% compared to wave one and wave two at 4.4% and 2.2%, respectively (Fig 3). Relative to wave two, residents in wave one had increased odds of developing delirium (OR 2.06; 95% CI 1.80–2.37). Residents in the off-peak period had even greater odds of delirium development (OR 5.54; 95% CI 4.96–6.19).

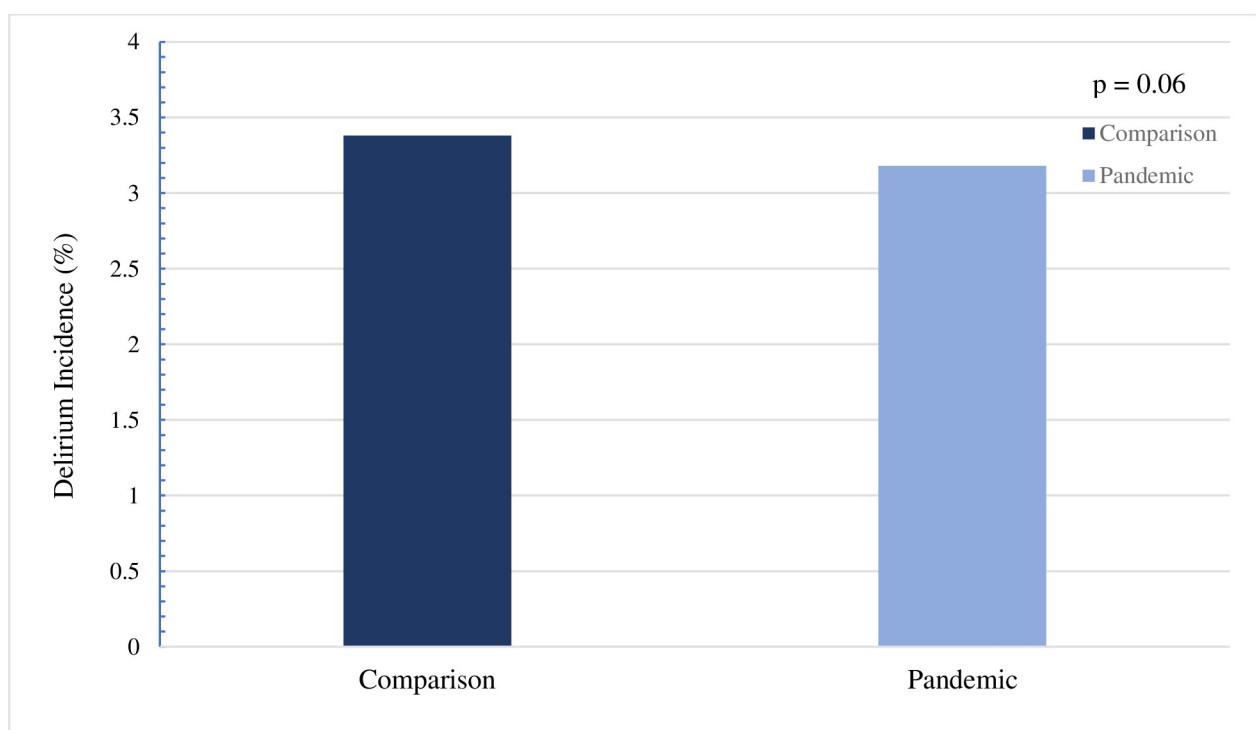

**Fig 2. Delirium incidence in comparison and pandemic samples in Ontario LTC homes.** Delirium incidence was calculated by assessing whether the delirium clinical assessment protocol was triggered in any follow-up assessment from baseline. The number of residents with triggered assessments was divided by the total number of residents in the sample, therefore displayed values are in percentage. A $\chi^2$ statistic was used to determine whether there was a statistically significant difference between the samples.

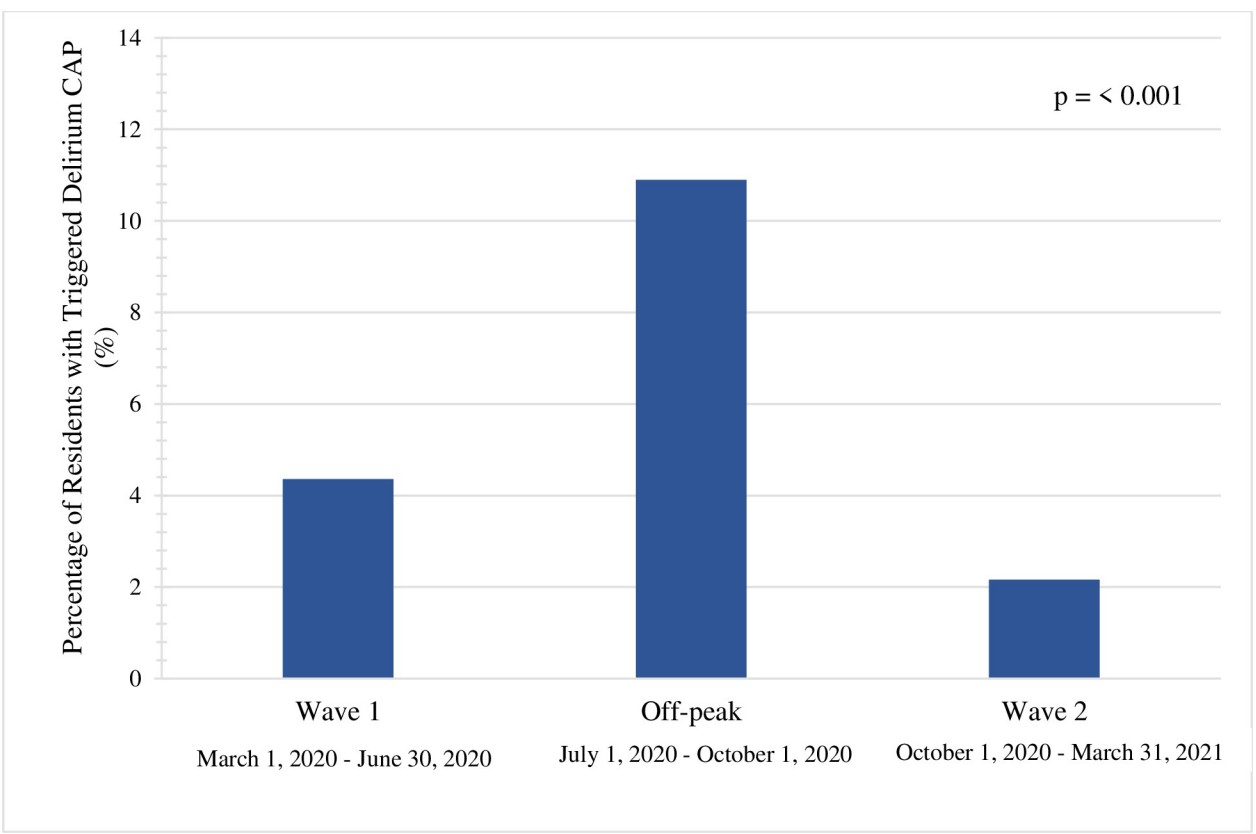

**Fig 3. Percentage of residents who trigger the delirium clinical assessment protocol within each pandemic wave.** Within the pandemic sample, residents who triggered the delirium clinical assessment protocol were further subdivided into waves, based on the date. The subsequent dates are included, with wave 1 being from March 1, 2020 to June 30, 2020. The off-peak period was considered between July 1, 2020 to September 30, 2020, with wave 2 being from October 1, 2020 to March 31, 2021. The number of residents who triggered the delirium clinical assessment protocol was divided by the total number of residents within that wave. Values are displayed in percentage, with a $\chi^2$ statistic used to determine whether there was any statistically significant difference.

Table 1 presents the baseline demographic and clinical characteristics of residents in the comparison and pandemic samples. There were slight differences in age, ADL dependence, cognitive impairment, and frailty between the samples. The pandemic sample had a slightly younger population, with a higher percentage of those with moderate cognitive impairment. There was also an increased proportion of those with advancing ADL dependence and a higher degree of frailty. There was a decreased proportion of new admissions in the pandemic sample. Within this, the pandemic sample had an increased percentage of admissions from the hospital setting, with a decreased percentage of admissions coming from the community or other inpatient facilities.

The results of the univariate analysis are displayed in Table 2. There was no significant association between the pandemic and delirium development. Increased odds of developing delirium were associated with individuals who were older, hearing impaired, cognitively impaired, frail, and health instable. A diagnosis of anxiety, bipolar disorder, or dementia, including Alzheimer's, showed increased odds of delirium development. An increasing number of medications used, a new medication being used within the last 90 days, and usage of analgesics, anxiolytics, antidepressants, and diuretics were all associated with higher odds of delirium development. Decreased odds of developing delirium were found in individuals who were newly admitted, experienced increasing difficulty with ADLs, and who had a recent hospital

**Table 1. Baseline demographic and clinical characteristics of residents in comparison and pandemic samples.**

| Characteristics | Comparison Sample n = 64,704 (%) | Pandemic Sample n = 55,370 (%) | P-Value |
|---|---|---|---|
| Age, y | | | |
| 18–64 | 4327 (6.7) | 3967 (7.2) | |
| 65–74 | 7936 (12.3) | 7300 (13.2) | |
| 75–84 | 17193 (26.6) | 14641 (26.4) | |
| 85+ | 35248 (54.5) | 29462 (53.2) | < 0.001 |
| Sex, female | 42940 (66.4) | 36597 (66.1) | 0.33 |
| Marital Status, married | 15454 (23.9) | 12808 (23.1) | 0.002 |
| Dementia (including Alzheimer's) | 35283 (54.5) | 29917 (54.0) | 0.08 |
| Hearing Impairment | | | |
| 0 (none) | 41186 (63.6) | 35736 (64.5) | |
| 1 (mild) | 16069 (24.8) | 13494 (24.4) | |
| 2–3 (moderate to severe) | 7449 (11.5) | 6140 (11.1) | 0.004 |
| Vision Impairment | | | |
| 0 (none) | 41579 (64.3) | 35965 (64.9) | |
| 1–2 (mild to moderate) | 20662 (31.9) | 17396 (31.4) | |
| 3–4 (severe) | 2463 (3.8) | 2009 (3.6) | 0.026 |
| Activities of Daily Living Hierarchy Scale | | | |
| 0–1 (no to mild difficulty) | 6399 (9.9) | 5045 (9.1) | |
| 2–3 (mild to moderate difficulty) | 29449 (45.5) | 23262 (42.0) | |
| 4–5 (moderate to extensive difficulty) | 27774 (42.9) | 25976 (46.9) | |
| 6 (total dependence) | 1082 (1.7) | 1087 (2.0) | < 0.001 |
| Changes in End-Stage Signs and Symptoms Scale | | | |
| 0 (no instability) | 31233 (48.3) | 26712 (48.2) | |
| 1–2 (mild instability) | 31923 (49.3) | 27357 (49.4) | |
| 3+ (moderate to severe instability) | 1548 (2.4) | 1301 (2.3) | 0.88 |
| Cognitive Performance Scale | | | |
| 0 (no impairment) | 7378 (11.4) | 6208 (11.2) | |
| 1–2 (mild impairment) | 23317 (36.0) | 19127 (34.5) | |
| 3 (moderate impairment) | 34009 (52.6) | 30035 (54.2) | < 0.001 |
| Frailty Index | | | |
| < 0.3 (no to mild frailty) | 2667 (4.1) | 1983 (3.6) | |
| 0.3–0.39 | 12727 (19.7) | 10114 (18.3) | |
| 0.4–0.49 | 23324 (36.0) | 19529 (35.3) | |
| 0.5–0.59 | 21038 (32.5) | 18972 (34.3) | |
| ≥ 0.6 (severe frailty) | 4948 (7.6) | 4772 (8.6) | < 0.001 |
| Dehydration | 4184 (6.5) | 3756 (6.8) | 0.027 |
| Weight Loss | 3299 (5.1) | 3531 (6.4) | < 0.001 |
| Anxiety Disorder | 9510 (14.7) | 8666 (15.6) | < 0.001 |
| Bipolar Disorder | 1836 (2.8) | 1756 (3.2) | < 0.001 |
| Depression | 20768 (32.1) | 18469 (33.4) | < 0.001 |
| Schizophrenia | 2737 (4.2) | 2559 (4.6) | 0.001 |
| Number of Medications | | | |
| 0–5 | 7531 (11.6) | 5994 (10.8) | |
| 6–9 | 19644 (30.4) | 16239 (29.3) | |
| 10+ | 37529 (58.0) | 33137 (59.8) | < 0.001 |
| Analgesic | 41440 (64.0) | 36688 (66.3) | < 0.001 |
| Antidepressant | 35714 (55.2) | 31902 (57.6) | < 0.001 |

(*Continued*)

**Table 1.** (Continued)

| Characteristics | Comparison Sample n = 64,704 (%) | Pandemic Sample n = 55,370 (%) | P-Value |
|---|---|---|---|
| Antipsychotic | 14806 (22.9) | 13583 (24.5) | < 0.001 |
| Anxiolytic | 6222 (9.6) | 5284 (9.5) | 0.67 |
| Diuretic | 18190 (28.1) | 15437 (27.9) | 0.37 |
| Hypnotic | 2549 (3.9) | 2172 (3.9) | 0.88 |
| New medication in last 90 days | 25202 (38.9) | 21142 (38.2) | 0.006 |
| Recent Hospital Stay | 9771 (15.1) | 8846 (16.0) | < 0.001 |
| New Admission | 20169 (31.2) | 11916 (21.5) | < 0.001 |
| Type of New Admission | | | |
| Hospital | 4572 (7.1) | 4960 (9.0) | |
| Community | 8625 (13.3) | 3434 (6.2) | |
| Other Inpatient | 6972 (10.8) | 3522 (6.4) | |
| Not New | 44535 (68.8) | 43454 (78.5) | < 0.001 |

stay. The interaction terms between the independent variables and the pandemic term were not significant and did not remain in our models.

The results of the final multivariate models are displayed in Table 2. Individuals who were older, cognitively impaired, and increasingly frail had greater odds of developing delirium. Moderate to severe health instability, a diagnosis of bipolar disorder, and use of an anxiolytic within the last seven days were also factors that increased the odds of developing delirium. Individuals who were newly admitted and those with moderate difficulty to total dependence with ADLs had lower odds of delirium development. The CHESS scale includes assessment of decline in cognition as part of determining an individual's level of frailty-related health instability. To ensure against multicollinearity, a sensitivity analysis was subsequently run with the CHESS scale removed. The results of the final multivariate models with the CHESS scale removed demonstrated no difference.

## Discussion

Our study found that the incidence of delirium was not different between the year before and the first year of the pandemic in Ontario LTC homes. During the pandemic, residents in the off-peak period were found to have the highest incidence of delirium and had the greatest odds of developing delirium. Clinical factors that increased delirium incidence included advancing age, frailty, cognitive impairment, and health instability. Residents who were newly admitted or had increasing difficulty with ADLs had lower odds of developing delirium. By assessing these clinical factors, preventative strategies can be targeted to those residents who benefit most. Additionally, finding no difference in delirium incidence during the pandemic may suggest that preventative strategies employed by LTC homes during COVID-19 were effective, and could be beneficial in the case of a future pandemic.

We did not find a statistically significant difference between the delirium incidence of the two samples. The pandemic did not increase delirium development. As mentioned previously, research regarding delirium in the LTC setting during the pandemic has not been abundant. Because we did not find a difference in delirium incidence, this raises two possible suggestions.

The first potential explanation for our results is that LTC homes were indeed able to mitigate the effects of the pandemic on their residents. Within the media, there was a substantial increase in the volume of negative stories broadcast regarding LTC during the pandemic [40]. This negative media portrayal of LTC would lead most to believe an increase in the incidence

**Table 2. Univariate and multivariate model results.**

| Characteristics | Univariate | Multivariable |
|---|---|---|
| | OR (95% CI) | OR (95% CI) |
| Pandemic (REF = no pandemic) | 0.94 (0.88–1.00) | |
| Age, y | | |
| 18–64 | REF | REF |
| 65–74 | 1.18 (1.00–1.39) | 1.17 (0.99–1.39) |
| 75–84 | 1.26 (1.08–1.46) | 1.29 (1.11–1.51) |
| 85+ | 1.35 (1.17–1.56) | 1.41 (1.22–1.64) |
| Sex (REF = female) | 0.91 (0.85–0.98) | |
| Marital Status, married (REF = not married) | 0.96 (0.89–1.04) | |
| Dementia (including Alzheimer's) (REF = no dementia) | 1.08 (1.01–1.15) | |
| Hearing Impairment | | |
| 0 (none) | REF | |
| 1 (mild) | 1.13 (1.05–1.22) | |
| 2–3 (moderate to severe) | 1.11 (1.01–1.23) | |
| Vision Impairment | | |
| 0 (none) | REF | |
| 1–2 (mild to moderate) | 0.99 (0.92–1.06) | |
| 3–4 (severe) | 1.14 (0.97–1.34) | |
| Activities of Daily Living Hierarchy Scale | | |
| 0–1 (no to mild difficulty) | REF | REF |
| 2–3 (mild to moderate difficulty) | 0.98 (0.88–1.10) | 0.88 (0.78–1.00) |
| 4–5 (moderate to extensive difficulty) | 0.78 (0.70–0.87) | 0.63 (0.55–0.72) |
| 6 (total dependence) | 0.56 (0.40–0.77) | 0.46 (0.33–0.64) |
| Changes in End-Stage Signs and Symptoms Scale | | |
| 0 (no instability) | REF | REF |
| 1–2 (mild instability) | 1.07 (1.00–1.14) | 1.02 (0.96–1.10) |
| 3+ (moderate to severe instability) | 1.60 (1.33–1.92) | 1.44 (1.19–1.74) |
| Cognitive Performance Scale | | |
| 0 (no impairment) | REF | REF |
| 1–2 (mild impairment) | 1.47 (1.30–1.66) | 1.45 (1.28–1.64) |
| 3 (moderate impairment) | 1.32 (1.17–1.49) | 1.24 (1.09–1.40) |
| Frailty Index | | |
| < 0.3 (no to mild frailty) | REF | REF |
| 0.3–0.39 | 0.89 (0.77–1.04) | 0.97 (0.83–1.14) |
| 0.4–0.49 | 0.94 (0.82–1.08) | 1.10 (0.94–1.28) |
| 0.5–0.59 | 1.24 (1.08–1.43) | 1.47 (1.26–1.73) |
| $\geq$ 0.6 (severe frailty) | 1.37 (1.14–1.65) | 1.61 (1.31–1.98) |
| Dehydration (REF = no dehydration) | 1.05 (0.92–1.19) | |
| Weight Loss (REF = no weight loss) | 0.93 (0.80–1.07) | |
| Anxiety Disorder (REF = no anxiety disorder) | 1.09 (1.00–1.18) | |
| Bipolar Disorder (REF = no bipolar disorder) | 1.30 (1.10–1.54) | 1.27 (1.07–1.51) |
| Depression (REF = no depression) | 1.06 (0.99–1.14) | |
| Schizophrenia (REF = no schizophrenia) | 1.06 (0.91–1.23) | |
| Number of Medications | | |
| 0–5 | REF | |
| 6–9 | 1.18 (1.04–1.33) | |
| 10+ | 1.28 (1.15–1.44) | |

(*Continued*)

**Table 2.** (Continued)

| Characteristics | Univariate | Multivariable |
|---|---|---|
| | OR (95% CI) | OR (95% CI) |
| Analgesic (REF = no analgesic) | 1.10 (1.03–1.18) | |
| Antidepressant (REF = no antidepressant) | 1.17 (1.10–1.25) | |
| Antipsychotic (REF = no antipsychotic) | 1.06 (0.98–1.14) | |
| Anxiolytic (REF = no anxiolytic) | 1.18 (1.07–1.31) | 1.12 (1.01–1.25) |
| Diuretic (REF = no diuretic) | 1.11 (1.03–1.19) | |
| Hypnotic (REF = no hypnotic) | 1.09 (0.93–1.28) | |
| New Medication (REF = no new medications) | 1.07 (1.00–1.14) | |
| Recent Hospital Stay (REF = no recent hospital stay) | 0.84 (0.77–0.93) | |
| New Admission (REF = not a new admission) | 0.64 (0.59–0.70) | 0.65 (0.60–0.71) |
| Type of New Admission | | |
| Hospital | REF | |
| Community | 1.10 (0.92–1.32) | |
| Other Inpatient | 1.08 (0.90–1.30) | |

of delirium would be seen due to the drastic environmental modifications put into place to prevent infection. It has been suggested that these isolation protocols and resource shortages would present increased barriers to facilitating evidence-based delirium prevention techniques [41]. However, as we did not see a difference in delirium incidence during the pandemic, this may support the idea that staff were still able to provide care that allowed for reorientation, and other preventative strategies to be used. Additional research is required to examine the individual structure of each LTC home and if or how these preventative strategies were able to continue to be used throughout the pandemic.

An alternative explanation for our results is the possibility that the true incidence of delirium during the pandemic was underestimated in the study homes. It has been hypothesized that because delirium is a presenting clinical symptom of COVID-19 infection, there would be a subsequent increased incidence of delirium during the pandemic period [42]. However, our results did not show this. Further research would be pertinent to examine delirium rates in a LTC population of residents known to be infected with COVID-19 compared with those who were not. For the purposes of this study, we did not have access to any data regarding whether a LTC home experienced a COVID-19 outbreak, nor how they managed it. Therefore, the scope of this study did not extend to further investigate this variable. Those that experienced COVID-19 outbreaks may very well have had additional resource shortages that posed a barrier to clinically identifying the signs of delirium in their residents when attention was focused on controlling spread of disease. Assessing these characteristics would be extremely informative in future studies to obtain a clearer picture.

In addition, infection with COVID-19 led to a great number of resident deaths. This was not accounted for by our study, and therefore our results may reflect a lower incidence as it does not consider delirium developed directly prior to resident deaths. That said, in the first wave of COVID-19 when most LTC deaths occurred, the median infection rate was 4.6% of residents per home. 15% of homes had severe outbreaks where 20% or more residents died, and just 5% of LTC homes accounted for over 54% of resident deaths [43]. In that light, it may be the case that a multi-level analysis adjusting for LTC home level covariates may be useful for future research.

Relative to wave two of the pandemic, individuals had greater odds of developing delirium during wave one or the off-peak period of the pandemic. Research conducted in Ontario LTC

homes found that residents had higher infection rates of COVID-19 in wave one compared to wave two [44]. Because delirium could be a presenting symptom of COVID-19 infection in this population, this could explain the higher odds of delirium development in this first wave relative to the second. However, the off-peak period showed even greater odds of delirium development, where incidence of COVID-19 infection in Ontario LTC residents was much lower during this period [44]. These increased odds during the off-peak period could reflect a delayed response to the infection control measures put in place in LTC homes. Beginning in March and continuing into the off-peak period, there was restriction of visitors, family members, and volunteers. Additional restrictions were put into place to limit resident contact within LTC homes, including for dining, group activities, and walks [44]. As more restrictions were put into place, spanning into the off-peak period, this could potentially explain the increased odds for delirium development during this time. As these restrictions began to lift later in the pandemic, we saw a decreased percentage of residents developing delirium moving into wave two. Although there was an increase in COVID-19 infection incidence in LTC in wave two, this incidence was less than that found during wave one [44].

Factors that were found to be associated with increased odds of developing delirium, including age greater than 74 years, cognitive impairment, moderate to severe health instability, and increasing levels of frailty were all aligned with previous literature [2, 8, 36, 45]. Individuals who were newly admitted during the selected time frame were found to have decreased odds of developing delirium. No studies have been conducted specifically examining LTC admission status related to delirium development. Much of the literature reflects the idea that delirium is an underrecognized condition [35, 45, 46]. Because delirium is an acute change from the patient's baseline mental status, to make the diagnosis the examiner needs to have a good working knowledge of the patient's baseline and how acutely these changes occurred [46]. It is possible that our results reflect under recognition of delirium in this population due to decreased familiarity between the newly admitted individual and staff conducting the assessment. On the contrary, because our study excluded residents with baseline delirium, these results could indicate better detection of delirium upon admission. Due to the study's design, this cannot be concluded with certainty. Future studies could look at comparison of delirium incidence in populations newly admitted to LTC compared to those with a longstanding residence.

Individuals with worse ADL impairment were found to have decreased odds of delirium development. Delirium can present as hyperactive, hypoactive, or a mixture of both. Hypoactive delirium often presents with lethargy, sedation, and slowed movements [3]. Previous studies have found that individuals with a predominantly hypoactive form of delirium often go unrecognized [20]. If an individual is dependent on nursing staff for locomotion, eating, toileting, and personal hygiene, a hypoactive delirium causing lethargy and slowed movements interlaid on top may not be recognized. Therefore, these results may reflect the under recognition of delirium in this population. A secondary hypothesis could be that individuals with increasing ADL dependence would subsequently have more frequent contact with nursing staff. This increased frequency of contact allows for more opportunities for reorientation, and relatively less isolation; thus, providing protection against delirium. Future studies should examine the mechanisms related to the incidence of delirium in residents with more ADL impairment.

A strength of this study is the large sample size we were able to obtain in the comparison and pandemic samples. The data obtained from MDS 2.0 was able to provide comprehensive information on LTC residents for the 13 months prior to the start of the COVID-19 pandemic, and then the succeeding 13 months. This allowed us to examine whether there was any effect of the pandemic, as well as assess several factors for associations.

There were a number of limitations that can be noted in this study. Features of the LTC homes related to the pandemic were not assessed. The amount of crowding, for-profit status, and staff incidence of COVID-19 are all risk factors that have been identified to determine whether a LTC home will experience an outbreak of COVID-19 and its intensity [44]. Looking at the specific infection control protocols put into place or identifying homes with COVID-19 outbreaks may have affected delirium incidence in those specific homes but was beyond the scope of this study. Additionally, death was not considered a competing factor within the study. Residents may have died prior to follow-up assessments. Further, delirium may have occurred prior to death, which would not have been accounted for as those residents receiving hospice care or those with end stage disease were excluded. As noted previously, this may have caused us to underestimate delirium incidence. Information about the length and severity of delirium was not available and thus could not be included.

## Conclusion

Our study found that the incidence of delirium was similar in the year prior to and the first year of the COVID-19 pandemic in Ontario LTC homes, suggesting that LTC homes were able to mitigate the effects of the pandemic. Future research would be beneficial to identify particular delirium preventative strategies and programs that were used within specific homes during the pandemic. Interventions to prevent delirium may benefit LTC residents who are older, living with cognitive impairment, health instability, or frailty as they are at an increased risk. LTC residents who are newly admitted to the home or dependent for their ADLs may be at risk of having delirium go unrecognized, however, further research is necessary to fully investigate this concept.

## Author Contributions

**Conceptualization:** Lydia Kennedy, George Heckman, Samuel D. Searle, Caitlin McArthur.

**Data curation:** John P. Hirdes.

**Formal analysis:** Lydia Kennedy, Caitlin McArthur.

**Funding acquisition:** John P. Hirdes.

**Investigation:** Lydia Kennedy, Caitlin McArthur.

**Methodology:** Lydia Kennedy, John P. Hirdes, George Heckman, Samuel D. Searle, Caitlin McArthur.

**Project administration:** Caitlin McArthur.

**Supervision:** John P. Hirdes, George Heckman, Caitlin McArthur.

**Validation:** Lydia Kennedy, Caitlin McArthur.

**Visualization:** Lydia Kennedy, Caitlin McArthur.

**Writing – original draft:** Lydia Kennedy.

**Writing – review & editing:** Lydia Kennedy, John P. Hirdes, George Heckman, Samuel D. Searle, Caitlin McArthur.

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
