## [Decision Letter · Decision Letter 0]

10 Jul 2024

PONE-D-24-17753The effect of the COVID-19 pandemic on delirium incidence in Ontario long-term care homes: A retrospective cohort studyPLOS ONE

Dear Dr. McArthur,

Thank you for submitting your manuscript to PLOS ONE. After careful consideration, we feel that it has merit but does not fully meet PLOS ONE’s publication criteria as it currently stands. Therefore, we invite you to submit a revised version of the manuscript that addresses the points raised during the review process.

 In particular, reviewers identified that the risk of bias in the results was very high.  Please submit your revised manuscript by Aug 24 2024 11:59PM. If you will need more time than this to complete your revisions, please reply to this message or contact the journal office at plosone@plos.org. Please include the following items when submitting your revised manuscript:A rebuttal letter that responds to each point raised by the academic editor and reviewer(s). You should upload this letter as a separate file labeled 'Response to Reviewers'.A marked-up copy of your manuscript that highlights changes made to the original version. You should upload this as a separate file labeled 'Revised Manuscript with Track Changes'.An unmarked version of your revised paper without tracked changes. You should upload this as a separate file labeled 'Manuscript'.

We look forward to receiving your revised manuscript.

Kind regards,

Kathleen Bennett

Academic Editor

PLOS ONE

Journal Requirements:

2. Thank you for stating in your Funding Statement: "The study is supported in part by funding from the Government of Canada's New Frontiers in Research Fund (NFRF; NFRFG-2020-00500) for collaboration in the EU Horizon 2020 research and innovation project Individualized CARE for Older Persons with Complex Chronic Conditions in Home Care and Nursing Homes (I-CARE4OLD, Grant Agreement No 965341). In addition, it received funding through the Canadian Institutes of Health Research (CIHR Reference # GA6-177780 )."

3. Thank you for stating the following financial disclosure: The study is supported in part by funding from the Government of Canada's New Frontiers in Research Fund (NFRF; NFRFG-2020-00500) for collaboration in the EU Horizon 2020 research and innovation project Individualized CARE for Older Persons with Complex Chronic Conditions in Home Care and Nursing Homes (I-CARE4OLD, Grant Agreement No 965341). In addition, it received funding through the Canadian Institutes of Health Research (CIHR Reference # GA6-177780 )."

Reviewers' comments:

Reviewer's Responses to Questions

**Comments to the Author**

1. Is the manuscript technically sound, and do the data support the conclusions?

Reviewer #1: Yes

Reviewer #2: Yes

2. Has the statistical analysis been performed appropriately and rigorously? 

Reviewer #1: Yes

Reviewer #2: Yes

3. Have the authors made all data underlying the findings in their manuscript fully available?

Reviewer #1: No

Reviewer #2: Yes

4. Is the manuscript presented in an intelligible fashion and written in standard English?

Reviewer #1: Yes

Reviewer #2: Yes

5. Review Comments to the Author

Reviewer #1: The manuscript is well written and informative. The data presented are relevant for clinical practice and policy makers, but there are some major issues needing attention from the authors.

First, delirium underdetection was extremely probable in the context of pandemic waves, because clinical resources were engaged into the care of COVID-19 cases and prevention of infection spread. The circumstance that the incidence of delirium was reduced during pandemic waves and raised in the inter-wave period supports this assumption. Therefore, the riskof bias in the results of this manuscript is very high.

Furthermore, were any of the participating nursing homes involved in COVID-19 outbreaks? Considering the characteristics of SARS-CoV-2 infection during the first and the second wave, this occurrence is extremely probable. How were these outbreaks managed? Did they affect the delirium incidence? In older patients, delirium is one of the possible clinical presentations of COVID-19, so an increase in the delirium occurrence during pandemic waves should have been expected. But surprisingly, the incidence of delirium was higher during the inter-peak period.

It is also unclear if some patients who had a long nursing home stay were included in both the pre-pandemic and the pandemic groups. How were these cases, that stayed in nursing homes from 2019 to 2021, managed?

Frailty index calculation was also based on clinical data collected not always during the periods of observation, but in some cases at the moment of admission into nursing homes, which may have substantially preceeded the actual period of observation. Therefore, since frailty is a dynamic process, the present analysis may have been based on inactual measurements of frailty.

Reviewer #2: The article is well written, with clear tables and figures. Furthermore it provides great insights in delirium incidence in Ontario, Canada. It gives very interesting insights and results, that are thoroughly discussed.

Figure 1 is difficult to read and should be uploaded in better quality.

6. PLOS authors have the option to publish the peer review history of their article (what does this mean?). If published, this will include your full peer review and any attached files.

Reviewer #1: No

Reviewer #2: **Yes: **Kelly Sabbe

---

## [Author Response · Author response to Decision Letter 0]

19 Aug 2024

Reviewer #1

Review Comments to the Author

The manuscript is well written and informative. The data presented are relevant for clinical practice and policy makers, but there are some major issues needing attention from the authors. 

Thank you for your suggested revisions to our manuscript. We have addressed these below, as well as in the attached revised manuscript. 

1. First, delirium under detection was extremely probable in the context of pandemic waves, because clinical resources were engaged into the care of COVID-19 cases and prevention of infection spread. The circumstance that the incidence of delirium was reduced during pandemic waves and raised in the inter-wave period supports this assumption. Therefore, the risk of bias in the results of this manuscript is very high.

We have revised the discussion to further address this risk of bias in regard to the comments above with the following lines.

“For the purposes of this study, we did not have access to any data regarding whether a LTC home experienced a COVID-19 outbreak, nor how they managed it. Therefore, the scope of this study did not extend to further investigate this variable. Those that experienced COVID-19 outbreaks may very well have had additional resource shortages that posted a barrier to clinically identifying the signs of delirium in their residents when attention was focused on controlling spread of disease. Assessing these characteristics would be extremely informative in future studies to obtain a clearer picture.” (Lines 347-353)

2. Furthermore, were any of the participating nursing homes involved in COVID-19 outbreaks? Considering the characteristics of SARS-CoV-2 infection during the first and the second wave, this occurrence is extremely probable. How were these outbreaks managed? Did they affect the delirium incidence? In older patients, delirium is one of the possible clinical presentations of COVID-19, so an increase in the delirium occurrence during pandemic waves should have been expected. But surprisingly, the incidence of delirium was higher during the inter-peak period.

We have revised the discussion to address the topic of assessing LTC homes with COVID-19 outbreaks. Identifying and analyzing these homes was beyond the scope of this study but has now been explicitly stated within the discussion and suggested for a future point of study. The following lines have been included:

“Further research would be pertinent to examine delirium rates in a LTC population of residents known to be infected with COVID-19 compared with those who were not. For the purposes of this study, we did not have access to any data regarding whether a LTC home experienced a COVID-19 outbreak, nor how they managed it.” (Lines 345-348)

“In addition, infection with COVID-19 led to a great number of resident deaths. This was not accounted for by our study, and therefore our results may reflect a lower incidence as it does not consider delirium developed directly prior to resident deaths. That said, in the first wave of COVID-19 when most LTC deaths occurred, the median infection rate was 4.6% of residents per home. 15% of homes had severe outbreaks where 20% or more residents died, and just 5% of LTC homes accounted for over 54% of resident deaths [43]. In that light, it may be the case that a multi-level analysis adjusting for LTC home level covariates may be useful for future research.” (Lines 355-361)

3. It is also unclear if some patients who had a long nursing home stay were included in both the pre-pandemic and the pandemic groups. How were these cases, that stayed in nursing homes from 2019 to 2021, managed?

It is certainly possible that LTC residents who had a long nursing home stay are included within both the pre-pandemic and pandemic groups. The main objective of our study was to characterize delirium incidence and see how this was affected by the emergence of the COVID-19 pandemic. Our interest was directed on the effect of time, versus the effect of clinical characteristics. Each group had patients with delirium at the baseline assessment deleted so that we could investigate development of new delirium. Therefore, if there was overlap in the residents between both groups, it was determined to not have a significant impact on the delirium incidence obtained. To specify this within the manuscript, the following lines were edited and added. 

“As the aim of this study was to investigate the development of new delirium from baseline to follow-up, residents with indicators of delirium on their baseline assessment were excluded. Residents who remained in LTC throughout the period of study between February 1, 2019, and March 31, 2021, may have been included in both the comparison and pandemic samples. However, as residents with baseline delirium were deleted in both samples, it was determined that this would not have a large effect on the final incidences obtained.” (Lines 148-154)

4. Frailty index calculation was also based on clinical data collected not always during the periods of observation, but in some cases at the moment of admission into nursing homes, which may have substantially preceded the actual period of observation. Therefore, since frailty is a dynamic process, the present analysis may have been based on unactual measurements of frailty.

As noted within the “Sample construction” section, quarterly assessments do not capture all of the variables used within the calculation for frailty index. This is because the quarterly assessments are used to track the status of a resident in between comprehensive assessments and focuses on items that are not expected to change acutely. In the case of our study, this clinical data only needed to be collected from prior assessments when the baseline assessment was a quarterly. Therefore, only a subset of each sample needed these variables brought forward. The variables brought forward also only represented a subset of the variables used for frailty index calculation. Finally, in order to bring those variables forward, the most recent comprehensive (annual or admission) assessment that had been completed in the preceding 12 months was used. The following lines have been edited and added in order to better clarify this. 

“Admission and annual assessments are comprehensive and include all of the variables required for our study. However, quarterly assessments are shorter to complete and do not include certain items that are not expected to change acutely. Therefore, for a subset of each sample where a quarterly assessment was used as a resident’s baseline, certain independent variables of interest were not carried forward. These were therefore obtained from the most recent admission or annual assessment completed in the preceding year. As noted, variables carried forward included those expected to remain fairly constant, for example, marital status, hearing/vision impairment, Alzheimer’s status, and a subset of variables used within the Frailty Index.” (Lines 129-136)

Reviewer #2

Review Comments to the Author

The article is well written, with clear tables and figures. Furthermore, it provides great insights in delirium incidence in Ontario, Canada. It gives very interesting insights and results, that are thoroughly discussed.

Thank you for your suggested revisions on our manuscript. We have addressed these below and in the revised manuscript. 

1. Figure 1 is difficult to read and should be uploaded in better quality.

Figure 1 has been reuploaded and reviewed using the PACE software to achieve a high quality figure for publication.

---

## [Editor Report · Decision Letter 1]

13 Sep 2024

The effect of the COVID-19 pandemic on delirium incidence in Ontario long-term care homes: A retrospective cohort study

PONE-D-24-17753R1

Dear Dr. McArthur,

We’re pleased to inform you that your manuscript has been judged scientifically suitable for publication and will be formally accepted for publication once it meets all outstanding technical requirements.

Kind regards,

Kathleen Bennett

Academic Editor

PLOS ONE
---

## [Editor Report · Acceptance letter]

5 Nov 2024

PONE-D-24-17753R1 

PLOS ONE

Dear Dr. McArthur, 

I'm pleased to inform you that your manuscript has been deemed suitable for publication in PLOS ONE. Congratulations! Your manuscript is now being handed over to our production team.

Kind regards, 

on behalf of

Professor Kathleen Bennett 

Academic Editor

PLOS ONE